# Impact of Angiogenesis- and Hypoxia-Associated Polymorphisms on Tumor Recurrence in Patients with Hepatocellular Carcinoma Undergoing Surgical Resection

**DOI:** 10.3390/cancers12123826

**Published:** 2020-12-18

**Authors:** Hannah Miller, Zoltan Czigany, Isabella Lurje, Sophie Reichelt, Jan Bednarsch, Pavel Strnad, Christian Trautwein, Christoph Roderburg, Frank Tacke, Nadine Therese Gaisa, Ruth Knüchel-Clarke, Ulf Peter Neumann, Georg Lurje

**Affiliations:** 1Charité–Universitätsmedizin Berlin, Department of Surgery, Campus Charité Mitte|Campus Virchow-Klinikum, 13353 Berlin, Germany; hannah.miller@charite.de (H.M.); sreichelt@ukaachen.de (S.R.); 2Department of Surgery and Transplantation, University Hospital Rheinisch-Westfälische Technische Hochschule (RWTH) Aachen, 52074 Aachen, Germany; zczigany@ukaachen.de (Z.C.); isabella.lurje@charite.de (I.L.); jbednarsch@ukaachen.de (J.B.); uneumann@ukaachen.de (U.P.N.); 3Charité–Universitätsmedizin Berlin, Department of Gastroenterology and Hepatology, Campus Charité Mitte|Campus Virchow-Klinikum, 13353 Berlin, Germany; christoph.roderburg@charite.de (C.R.); frank.tacke@charite.de (F.T.); 4Department of Internal Medicine III, University Hospital Rheinisch-Westfälische Technische Hochschule (RWTH) Aachen, 52074 Aachen, Germany; pstrnad@ukaachen.de (P.S.); ctrautwein@ukaachen.de (C.T.); 5Institute of Pathology, University Hospital Rheinisch-Westfälische Technische Hochschule (RWTH) Aachen, 52074 Aachen, Germany; ngaisa@ukaachen.de (N.T.G.); rknuechel-clarke@ukaachen.de (R.K.-C.)

**Keywords:** hepatocellular carcinoma, single-nucleotide polymorphism, IL8

## Abstract

**Simple Summary:**

Hepatocellular carcinoma remains a leading cause of cancer-related death and the most common primary hepatic malignancy in the Western hemisphere. Previous research found that angiogenesis-related cytokines and elevated levels of interleukin 8 and vascular endothelial growth factor (VEGF) shorten the expected time of survival. Moreover, factors of tumor angiogenesis- and hypoxia-driven signaling pathways are already associated with worse outcome in disease-free survival in several tumor entities. Our study investigates the prognosis of hepatocellular carcinoma patients based on a selection of ten different single-nucleotide polymorphisms from angiogenesis, carcinogenesis, and hypoxia pathways. Our study with 127 patients found supporting evidence that polymorphisms in angiogenesis-associated pathways corelate with disease-free survival and clinical outcome in patients with hepatocellular carcinoma.

**Abstract:**

Tumor angiogenesis plays a pivotal role in hepatocellular carcinoma (HCC) biology. Identifying molecular prognostic markers is critical to further improve treatment selection in these patients. The present study analyzed a subset of 10 germline polymorphisms involved in tumor angiogenesis pathways and their impact on prognosis in HCC patients undergoing partial hepatectomy in a curative intent. Formalin-fixed paraffin-embedded (FFPE) tissues were obtained from 127 HCC patients at a German primary care hospital. Genomic DNA was extracted, and genotyping was carried out using polymerase chain reaction (PCR)–restriction fragment length polymorphism-based protocols. Polymorphisms in interleukin-8 (IL-8) (rs4073; *p* = 0.047, log-rank test) and vascular endothelial growth factor (VEGF C + 936T) (rs3025039; *p* = 0.045, log-rank test) were significantly associated with disease-free survival (DFS). After adjusting for covariates in the multivariable model, IL-8 T-251A (rs4073) (adjusted *p* = 0.010) and a combination of “high-expression” variants of rs4073 and rs3025039 (adjusted *p* = 0.034) remained significantly associated with DFS. High-expression variants of IL-8 T-251A may serve as an independent molecular marker of prognosis in patients undergoing surgical resection for HCC. Assessment of the patients’ individual genetic risks may help to identify patient subgroups at high risk for recurrence following curative-intent surgery.

## 1. Introduction

Hepatocellular carcinoma (HCC) is the most common primary hepatic malignancy, and its mortality ranks fourth among solid tumors, behind carcinomas of the lung, colon, and the stomach [1,2]. As most HCCs develop in the background of chronic liver disease, liver transplantation is considered the optimal curative therapy because it treats both the tumor and the underlying liver disease [3,4,5,6]. In the context of donor allograft shortage, surgical resection has emerged as a viable treatment strategy even beyond early Barcelona Clinic Liver Cancer (BCLC) stages [3,7]. Despite continuous advances in patient selection, surgical technique, and medical treatments [5,6], the rate of local tumor recurrence exceeds 50% 4 years after partial hepatectomy [8,9]. Therefore, the implementation of prognostic molecular markers as an adjunct to traditional staging systems may not only be helpful in identifying patients prone to recurrence but can also aid patient-specific treatment selection.

In 1971, Judah Folkman introduced the hypothesis that angiogenesis, the formation of new blood vessels from endothelial precursors, is a prerequisite for the growth and progression of solid malignancies. Gaining access to the host vascular system and the generation of a tumor blood supply are rate-limiting steps in tumor growth and progression [10]. Among proangiogenic factors, vascular endothelial growth factor (VEGF), a sub-family of growth factors which is induced under hypoxic conditions through hypoxia-inducible factor (HIF)-1α promotor binding, plays a pivotal role in tumor angiogenesis [11,12]. The relevance of VEGF-dependent pathways for hepatocarcinogenesis was recently emphasized by the data from a phase 3 clinical trial, establishing the combination therapy of the VEGF inhibitor bevacizumab and the immune checkpoint inhibitor atezolizumab as the first-line systemic treatment in advanced HCC [13].

Additionally, tumors can sustain angiogenesis in a VEGF-independent manner [14]. As such, interleukin (IL)-8 (CXCL8) signaling preserves the angiogenic phenotype in HIF1-α-deficient colon cancer cells, indicating a critical role of IL-8 in tumor-associated angiogenesis, independent of VEGF [15]. Furthermore, elevated serum levels of IL-8 and single-nucleotide polymorphisms (SNPs) of VEGF and IL-8 are associated with shorter disease-free survival (DFS) and overall survival (OS) in HCC and other gastrointestinal malignancies [16,17,18,19,20].

Based on these data, we here employed a pathway-focused approach to investigate whether functional single-nucleotide polymorphisms of genes involved in angiogenesis, tumorigenesis, and hypoxia (Table 1) are associated with differences in clinical outcome in HCC patients undergoing partial hepatectomy in a curative intent.

## 2. Results

### 2.1. Patients Characteristics

Between 2010 and 2017, 127 patients with localized hepatocellular carcinoma were included in this study. Thirty-seven (37/127, 29.1%) patients were female and 90/127 (70.9%) were male; the median age was 67 years (15–89). The median body mass index (BMI) was 27 kg/m^2^ (16.8–41). Sixty-four (64/127, 50.4%) patients had histologically confirmed cirrhosis, 76/127 (59.8%) had a single tumor, and 51/127 (39.3%) had more than one HCC lesion. Average diameter of tumor nodules on final pathology was 68 mm (6–228 mm). Median alpha-fetoprotein (AFP) at time of operation was 147.9 µL/L. Seventy-seven (77/127, 60.6%) patients belonged to BCLC category 0/A and fifty (50/123, 39.4%) patients to category B/C. Clinico-pathological characteristics are presented in Table 2 and Table 3.

The median OS was 30.5 months (range 0–122 months), and the median DFS was 23 months (range 0–117 months). During the observation period, 74 (58.2%) patients died and 62 (48.8%) experienced tumor recurrence (Table 2). T-category 3 or 4 (*p* < 0.001), more than one tumor node (*p* < 0.001), BCLC stage B/C (*p* < 0.001), and the presence of hepatic cirrhosis (*p* = 0.012) were significantly associated with a shorter disease-free survival. T-category higher than 3 (*p* = 0.002), more than one tumor nodule (*p* = 0.010), a tumor diameter over 50 mm (*p* value = 0.035), male sex (*p* = 0.002), and the presence of hepatic cirrhosis (*p* = 0.005) were associated with shorter OS.

### 2.2. Analysis of IL-8 T-251A and Clinical Outcome

Genotyping of IL-8 T-251A was successful in 125 of 127 patients and was used for the subsequent analysis. Of these, sixty-two (62/125, 49.6%) patients recurred, while 63/125 (50.4%) patients did not show a recurrence within the observation period. An A/A genotype was identified in 14 (11.3%) patients, who showed a median DFS of median 20 (0–40.1) months compared to 65 patients with the T/T and 46 with the A/T allele with a median survival of 32 months (CI 18.34–5.8). IL-8 T-251A had a minor allele frequency of 11.3% and was analyzed in a codominant model. The unfavorable A/A allele was interpreted against A/T and T/T as favorable alleles (Table 4, Figure 1a). The *p*-value of the log-rank test was 0.047 (Table 4, Figure 1a). IL-8 T-251A was included in the multivariable analysis. IL-8 251 SNPs showed no significant association with OS.

### 2.3. Analysis of VEGF C + 936T and Clinical Outcome

The genotyping of VEGF C + 936T was successful in all 127 patients. During the observation period, 61/127 (48%) patients showed recurrence. Of these, 41 with allele C/C after 41 months (CI 5–77), 22 with C/T allele after 21 months (CI 8–4) and 5 with allele T/T after 6 months (CI 2–10), respectively. The *p*-value of the log-rank test was 0.023 (Table 4, Figure 1b). Due to the small group of patients with a T/T allele, we performed a dominant analysis (C/T and T/T grouped together) with a *p*-value of 0.045. VEGF was included in the multivariable analysis. OS showed no significant association with the VEGF SNP and was, therefore, not further analyzed.

### 2.4. Combined Analysis of VEGF C + 936T, IL-8 T-251A and Clinical Outcome

When VEGF C + 936T (adjusted *p*-value = 0.332) and IL-8-251 T > A (adjusted *p*-value = 0.010) were stratified by cirrhosis, T-category, more than one tumor node, and polymorphisms in IL8 remained significantly associated with DFS. VEGF C + 936T did not remain associated with DFS. Patients with zero unfavorable alleles (IL-8-251 AT/TT and VEGF +936 CC allele) had lower risk of tumor recurrence than patients with one unfavorable allele (RR 1.853 (1.045–3.284)) or two unfavorable genes (RR 4.910 (1.047–23.031), adjusted *p*-value 0.034) (Table 5, Figure 1c). The median disease-free survival was 20 months in the group with one unfavorable allele and was 7 months with two unfavorable alleles compared to 33 months in the favorable group.

### 2.5. Analysis of Other Tested Germline Polymorphisms

In total, 10 genes were tested; however, none of the further genes showed any relevant association with DFS or OS. The detailed data are presented in Table 4.

## 3. Discussion

In this study, we aimed to clarify the impact of angiogenesis- and hypoxia-associated germline polymorphisms on tumor recurrence in HCC patients who underwent partial hepatectomy. Here, we demonstrate that a proangiogenic and functional germline polymorphism of the IL-8 gene significantly correlates with DFS in HCC patients undergoing curative-intent surgery.

Despite recent advancements in the surgical treatment of HCC, the high rate of tumor recurrence results in an overall poor clinical prognosis [3]. Although great efforts have been made to optimize patient selection based on clinical and molecular parameters, data on how germline polymorphisms influence clinical outcomes after surgical resection for HCC are scarce [41]. The prominent role of tumor angiogenesis in HCC biology is reflected in the prominent arterial neo-vascularization of HCC as one of the hallmarks of malignant hepatocyte transformation [42]. Targeted agents such as sorafenib, a small-molecule multityrosine kinase inhibitor, have been employed in the treatment of advanced HCC [17,43,44], and more recently, the administration of atezolizumab, a checkpoint inhibitor, in combination with an anti-VEGF antibody (bevacizumab) showed promising results [13].

VEGF is one of the main regulators of tumor angiogenesis and is regulated by a large variety of transcription factors such as HIF-1, nuclear factor κB (NF-κB), and signal transducer and activator of transcription 3 (STAT3) [14]. Activation of the VEGF signaling pathway leads to endothelial cell proliferation, migration, and formation of new vessels [45,46]. While the level of circulating VEGF protein is increased in HCC patients and has been linked to poor oncological outcomes [45], the expression levels of VEGF protein also correlate directly with tumor size, metastasis, and poor prognosis in various malignancies [47]. Functional DNA sequence variations within the VEGF gene lead to altered serum levels and activity and, as such, also affect clinical outcomes in different types of malignant disease [48,49]. The VEGF C + 936T SNP has been associated with an increased susceptibility to breast, lung, and colon cancer as well as with a shorter disease-free survival in colon cancer [20,50]. In our study, VEGF C + 936T and the presence of the T/T or C/T alleles was associated with shorter DFS in our univariable analysis but did not reach significance in the multivariable analysis. The DFS was 20 months in individuals with T/T or C/T alleles, in contrast to C/C homozygote patients who had a median DFS of 30 months.

IL-8, a member of the CXC chemokine family, is a potent and VEGF-independent mediator of tumor angiogenesis. In tumor biology, IL-8 conveys direct effects on tumor cells, mediating the transition to a migratory, proliferative, or mesenchymal phenotype [51]. Furthermore, IL-8 modifies the tumor microenvironment, encouraging the recruitment of pro-tumorigenic immune cells, such as tumor-associated macrophages with an “M2-like” phenotype, ultimately leading to a positive feedback loop with increased IL-8 production, as well as recruitment of cancer-associated neutrophils, which, in turn, contribute to angiogenesis and invasion through the secretion of matrix metalloproteinase (MMP)-9 [52,53]. Furthermore, IL-8 conveys direct proangiogenic, chemotactic effects on endothelial cells via its receptors CXCR1 and -2 [51]. Most studies on IL-8 in HCC focus on the prognostic role of IL-8 expression and on the inhibition or activation of IL-8 in HCC models and cell lines [22,23,38,54,55]. As such, the findings of different groups have linked increased serum levels of IL-8 with tumor invasion, recurrence, and metastases [22,55]. Only scarce evidence is available on the prognostic role of IL-8 germline SNPs in HCC patients undergoing curative-intent surgery [56,57,58]. In 2000, Hull et al. identified the functional role of a common polymorphism IL-8 T-251A bp upstream of the IL-8 transcriptional start site, with over 50% of the United Kingdom population being heterozygous. Their experimental data suggested the association of the IL-8 T-251A A allele with increased IL-8 production [59]. Furthermore, high-expression variants of IL-8 T-251A (A/A) are linked to tumor recurrence and poor clinical outcome in various gastrointestinal malignancies, including colon and gastric cancer [18,20]. In patients with HCC, the homozygous IL-8-A/A allele was recently associated with a favorable clinical effect on transcatheter arterial chemoembolization (TACE) efficacy and lower levels of serum AFP [60]. Angiogenesis Liver CancEr 2 (ALICE-2), a retrospective multicenter study of 210 patients with advanced HCC, recently identified a panel of H1F-1α, VEGF-A, VEGF-C, VEGFR-1, and VEGFR-3 variants to be associated with better response to sorafenib treatment [29]. The present study demonstrated that patients with high-expression variants of IL-8 T-251A (A/A) developed significantly earlier tumor recurrence after partial hepatectomy with curative intent. On average, individuals with the homozygous minor allele A/A were recurrence-free for 20 months, compared to 32-month RFS in patients with at least one “protective” T allele (*p*-value: 0.034 in multivariable analysis).

The findings of this study should be interpreted in light of the potential limitations; nonetheless, this type of pilot study is an ideal forum for testing a novel hypothesis and generating data that need to be confirmed in a prospective study. First, our findings are based on a relatively small but homogeneous number of Central European HCC patients; and secondly, we examined only 10 genes within the angiogenesis pathway. While it is recognized that the observed associations and patterns require confirmation with an independent dataset, we have taken care to select candidate genes with a documented role in tumor angiogenesis that were also found to be associated with prognosis in previous studies [61].

## 4. Materials and Methods

### 4.1. Patients

Between 2010 and 2017, unrelated patients with localized HCC and no signs of systemic disease were treated with first-line surgical resection at the University Hospital of the Rheinisch-Westfälische Technische Hochschule (RWTH) Aachen (UH-RWTH) and were retrospectively included in this study. Clinico-pathological and follow-up data were obtained from a prospectively managed institutional database and analyzed retrospectively [62]. Clinical staging was performed according to the criteria of the International Union Against Cancer (UICC), BCLC, and Milan criteria. An experienced pathologist (N.T.G.) reviewed the histological specimens of the cohort (*n* = 127) to confirm the diagnosis. Patient samples were provided by the Institute of Pathology (UH-RWTH) and the institutional Biobank (RWTH-cBMB). The study was conducted in accordance with the requirements of the Institutional Review Board of the RWTH Aachen University (EK 360/15), the current version of the Declaration of Helsinki, and the guidelines for good clinical practice.

### 4.2. Selection of Candidate Polymorphisms

SNPs for molecular testing were selected based on our previous studies [20] and with a focus on tumor angiogenesis based on VEGF-dependent and -independent IL-8 pathways and the hypoxia-driven pathway. Besides well-documented function of the selected genes in the above-mentioned pathways, proof of a biological relevance of the respective SNPs was a necessary prerequisite. Additionally, the frequency of the polymorphism had to be high enough to allow a meaningful statistical analysis [20] (Table 1 and Table 6).

### 4.3. Genotyping

From 127 formalin-fixed paraffin-embedded (FFPE) HCC samples, genomic DNA was extracted using the QIAmp DNA Isolation Kit (Qiagen, CA, USA) according to the manufacturer’s protocol. DNA quality was determined photometrically (NanoDrop, Thermo Fisher, MA, USA). Samples were analyzed with the polymerase chain reaction–restriction fragment length polymorphism (PCR-RFLP) technique. After PCR amplification of probes using forward and reverse primers, the products were digested with restriction enzymes (New England Biolab, MA, USA). Reaction products were separated on a DNA Stain G (SERVA, Heidelberg, Germany) stained 4% agarose gel at 120 mV for 60 min and visualized under UV light using a GelDoc system (Bio-Rad Laboratories GmbH, Feldkirchen, Germany). See Table 6 for forward and reverse primers, annealing temperatures, and restriction enzymes.

### 4.4. Study Endpoints and Statistical Analysis

The primary endpoint of this study was DFS, calculated from the date of surgery to the date of the first recurrence. Patients who did not recur were censored at the time of death or at last follow-up. The secondary endpoint was OS, defined as the period from surgery to the date of death from any cause or the last contact if the patient was alive.

Categorical data are presented as numbers and percentages and were compared using the chi-squared test, Fisher’s exact test, or linear-by-linear association according to scale and number count. Data derived from continuous variables are presented as mean and standard deviation and comparisons between different time points were made using the Mann–Whitney U test. The association between each SNP and DFS and OS was examined using Kaplan–Meier curves and log-rank tests. For polymorphisms with a genotype frequency (homozygous minor allele) of <10%, the associations between genotypes and clinical outcome were analyzed in a dominant model. Otherwise, they were analyzed in a codominant or additive model.

Survival curves were generated using the Kaplan–Meier method and compared with the log-rank test. Univariable and multivariable associations of factors with patient survival and tumor recurrence were assessed using Cox proportional hazard models. Hazard ratios are presented with 95% confidence intervals (CI). Variables yielding significance in the univariable analysis were included in the multivariable analysis. The level of significance was set to *p* < 0.05. Analyses were performed using SPSS Statistics 23 (IBM Corp., Armonk, NY, USA).

## 5. Conclusions

Notwithstanding the aforementioned limitations, we have identified polymorphisms in IL-8 to be associated with tumor recurrence in HCC patients undergoing curative-intent surgery. Thus, the analysis of angiogenesis-related germline gene polymorphisms may facilitate more sophisticated patient selection by identifying patients at high risk for HCC recurrence. One might speculate that this subgroup of patients could potentially benefit from adjuvant anti-VEGF (e.g., bevacizumab) therapeutics. Despite these encouraging findings from our pilot study, independent, larger, controlled prospective biomarker-embedded clinical trials are warranted to validate our observations.

## Figures and Tables

**Figure 1 cancers-12-03826-f001:**
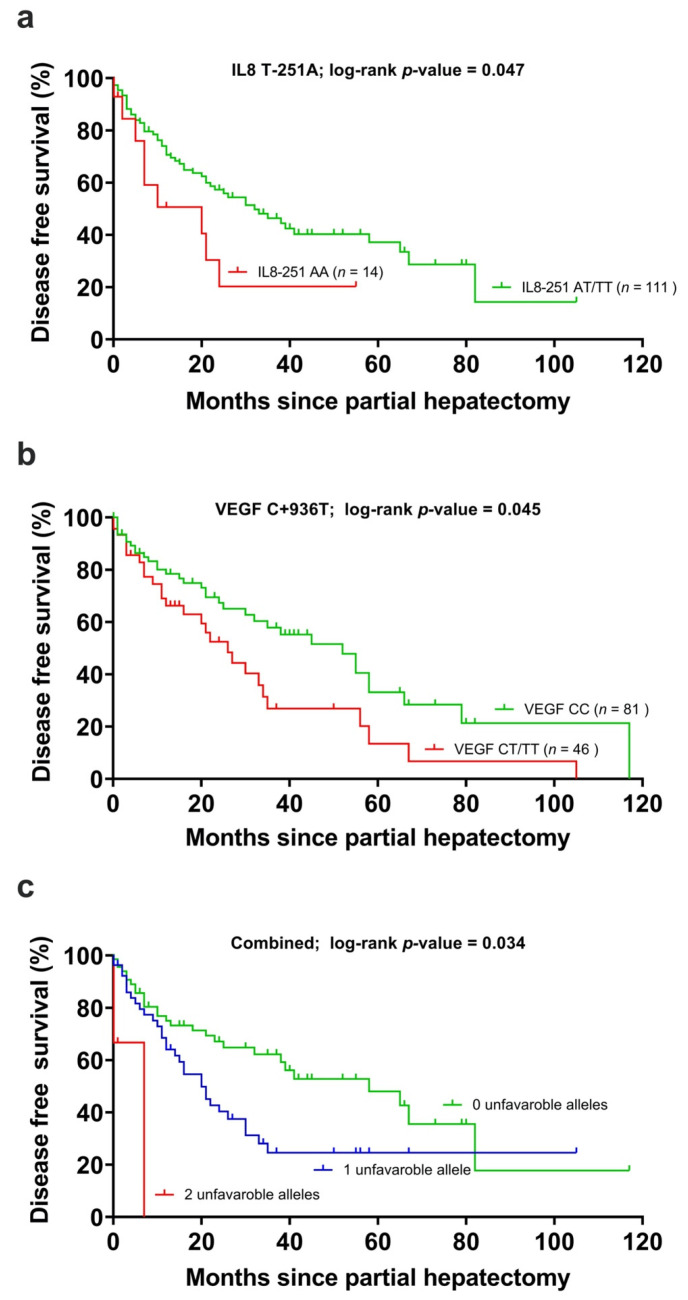
(**a**) Disease-free survival in the carriers of the interleukin-8 T-251A polymorphism, (**b**) the vascular endothelial growth factor C + 936T (VEGF) polymorphism and (**c**) the unfavorable alleles grouped together. Censored cases indicate the time of last follow-up for those patients who had neither recurred nor died at the time of the analysis of data.

**Table 1 cancers-12-03826-t001:** Genes, their function, and the minor allele frequency.

No.	Gene	Allele	SNP ^1^	Allele Function	Minor Allele Freq.	Description	Ref.
1	IL-8 − 251	A > T	rs4073	A-allele: higher IL-8 plasma levels	45%	The dominant allele AT and AA is associated with a poor prognosis and may increase CXCL 6, which stimulates endothelial production	[21,22,23,24,25]
2	VEGF + 936	C > T	rs3025039	T-allele: lower VEGF plasma levels	20%	De novo vascularization, endothelial proliferation Increased risk of NASH C-allele associated with higher VEGF production	[21,26,27,28]
3	EGFR-497	G > A	rs2227983	A-allele: lower EGFR ligand binding, growth stimulation	29%	Identified in gastrointestinal and colorectal tumors	[29,30]
4	EGFA 61G	A > G	rs4444903	A-allele: lower EGF serum levels	45%	Increased susceptibility to hepatitis C	[30,31,32]
5	p53	C > G	rs1042522	Tumor suppressor, induces cell cycle arrest	35%	Associates with LiFraumeni syndrome, is detected as inducer of different types of cancer	[33]
6	CXCR	G > C	rs2234671	C-allele: higher ligand binding	10%	Interleukin-8 acts through CXCR receptor	[34]
7	IL-1b	C > T	rs1143634	Increased tumor growth and proliferation	20%	Stomach cell cancer, gall bladder cancer, breast cancer, and liver cell cancer	[35]
8	HIF1 A588T	C > T	rs11549465	T-allele with cancer associated	8%	Associated with tumor size and location in colorectal carcinoma, tumor tissue is over-stimulated with HIF1	[36,37]
9	IL-10	T > G	rs1800872	Higher Il-10, more immunosuppressive effect	30–40%	Increases risk of liver cell cancer, negatively modulates NFKappaB and TNF alpha production (inhibits gene expression)	[38]
10	IL-6	G > C	rs1800795	Proinflammatory cytokine, response to cancer	20%	Increased risk of cervical and renal cancer Stimulates endothelial production	[39,40]

Abbreviations used: CI, confidence interval; VEGF, vascular endothelial growth factor; EGF, epidermal growth factor; EGFR, epidermal growth factor receptor; IL-1/6/8/10, interleukin-1/6/8/10; CXC2, chemokine receptor; HIFa, hypoxia-inducing factor alpha; NASH, non-alcoholic fatty liver disease; NF-κB, nuclear factor kappa-light-chain-enhancer of activated B cells; TNF-alpha, tumor-necrosis factor alpha; p53, protooncogene 53, ^1^ SNP, single nucleotide polymorphism Database NCBI.

**Table 2 cancers-12-03826-t002:** Demographic and clinico-pathological characteristics and disease-free survival.

Variable	Cutoff	Total No. Patients (%)	Event in Group (%)	Median DFS (Months) (95% CI)	Relative Risk (95% CI)	*p* Value ^1^
Sex	Female	37 (29.1%)	19 (51.4%)	38 (17.1–58.9)	0.763 (0.443–1.314)	0.329
	Male	90 (70.9%)	43 (47.8%)	26 (16.1–35.3)	1 (reference)	
Age (years)	<65	48 (37.8%)	26 (44.2%)	26 (12.7–39.3)	1.222 (0.736–2.032)	0.438
	>65	79 (62.2%)	36 (45.6%)	32 (17.0–47.0)	1 (reference)	
BMI	<25	51 (40.2%)	22 (43.1%)	35 (15.4–54.6)	0.790 (0.469–1.330)	0.375
	>25	76 (59.8%)	40 (52.6%)	26 (05.3–15.7)	1 (reference)	
Diameter (mm)	<50	49 (38.6%)	24 (49.0%)	39 (26.9–51.1)	0.695 (0.415–1.165)	0.168
	>50	78 (61.4%)	38 (48.7%)	21 (12.5–29.5)	1 (reference)	
T-category	T1/2	95 (74.8%)	51 (53.7%)	39 (12.1–65.9)	**0.407 (0.244–0.679)**	**<0.0001**
	T3/4	32 (25.2%)	12 (37.5%)	12 (05.1–18.9)	1 (reference)	
N-category	0	29 (22.8%)	13 (55.2%)	26 (05.4–47.0)	0.976 (0.543- 1.756)	0.936
	N	98 (74.7%)	41 (44.8%)	30 (12.9–47.1)	1 (reference)	
HCC tumor nodes	=1	51 (39.3%)	35 (68.6%)	12 (07.6–16.4)	**0.543 (0.342–0.862)**	**0.0001**
	>1	76 (59.8%)	27 (35.5%)	67 (43.3–90.7)	1 (reference)	
Steatosis	No	79 (61.9%)	39 (49.3%)	30 (18.2–41.8)	1.093 (0.652–1.833)	0.736
	Yes	48 (38.1%)	23 (47.9%)	25 (0.00–61.7)	1 (reference)	
Cirrhosis	No	63 (49.6%)	26 (41.3%)	58 (14.5–101.5)	**0.519 (0.308–0.874)**	**0.012**
	Yes	64 (50.4%)	36 (48.8%)	24 (15.7–32.3)	1 (reference)	
AFP (µL/L)	<100	50 (39.4%)	27 (54.0%)	32 (22.4–41.6)	0.732 (0.455–1.180)	0.200
	>100	77 (60.6%)	35 (55.5%)	24 (09.5–38.5)	1 (reference)	
BCLC Stage	0/A	77 (57.5%)	33 (42.8%)	65 (33.6–96.4)	**0.291 (0.171–0.496)**	**<0.0001**
	B/C	50 (39.4%)	33 (66.0%)	11 (06.7–15.2)	1 (reference)	

Values are given as median with 95% CI or numbers and (percentages). Results of the regression analysis are given as relative risk with 95% confidence interval. Bold is used to highlight significant results. ^1^ Based on log-rank test. BCLC—Barcelona Clinic Liver Cancer.

**Table 3 cancers-12-03826-t003:** Demographic and clinico-pathological characteristics and overall survival.

Variable	Cutoff	Total No. Patients (%)	Event in Group (%)	Median OS (Months) (95% CI)	Relative Risk (95% CI)	*p* Value ^2^
Sex	Female	37 (29.1%)	15 (40.5%)	78 (56.0–89.3)	**0.410 (0.231–0.723)**	**0.002**
	Male	90 (70.9%)	59 (65.6%)	24 (16.0–32.0)	1 (reference)	
Age (years)	<65	48 (37.8%)	27 (46.2%)	38 (00.0–77.8)	0.840 (0.523–1.351)	0.473
	>65	79 (62.2%)	47 (59.5%)	32 (18.7–45.3)	1 (reference)	
BMI	<25	51 (40.2%)	28 (54.9%)	31 (12.3–49.7)	0.9 (0.562–1.441)	0.660
	>25	76 (59.8%)	46 (60.5%)	35 (22.2–47.8)	1 (reference)	
Diameter (mm)	<50	49 (38.6%)	25 (51.0%)	58 (36.2–79.8)	**0.601 (0.370–0.976)**	**0.035**
	>50	78 (61.4%)	49 (62.8%)	23 (15.6–30.4)	1 (reference)	
T-category	T1/2	95 (74.8%)	47 (51.6%)	47 (26.8–67.2)	**0.261 (0.060–1.131)**	**0.002**
	T3/4	32 (25.2%)	24 (55.0%)	15 (06.4–23.6)	1 (reference)	
N-category	0	29 (22.8%)	14 (48.3%)	65 (26.5–103.5)	0.591 (0.324–1.077)	0.082
	N	98 (74.7%)	46 (59.0%)	32 (18.0–46.0)	1 (reference)	
HCC tumor nodes	=1	51 (39.3%)	38 (50.0%)	42 (22.9–61.1)	**0.543 (0.342–0.862)**	**0.010**
	>1	76 (59.8%)	35 (70.0%)	20 (12.0–28.0)	1 (reference)	
Steatosis	No	79 (61.9%)	51 (65.4%)	30 (18.2–29.2)	1.590 (0.963–2.623)	0.070
	Yes	48 (38.1%)	22 (45.8%)	58 (29.2–86.8)	1 (reference)	
Cirrhosis	No	63 (49.6%)	30 (47.6%)	55 (25.9–84.1)	0.515 (0.322–0.825)	0.005
	Yes	64 (50.4%)	44 (68.7%)	24 (11.9–36.1)	1 (reference)	
AFP (µL/L)	<100	50 (39.4%)	27 (54.0%)	42 (18.8–65.2)	1.025 (0.618–1.699)	0.924
	>100	77 (60.6%)	47 (61.0%)	29 (04.2–20.8)	1 (reference)	
BCLC Stage	0/A	77 (57.5%)	33 (42.8%)	47 (27.7–66.2)	**0.493 (0.309–0.789)**	**0.003**
	B/C	50 (39.4%)	33 (66.0%)	17 (09.1–24.9)	1 (reference)	

Values are given as median with 95% CI, numbers and (percentages). Results of the regression analyses are given as relative risk with 95% confidence interval. Bold is used to highlight significant results. ^2^ Based on log-rank test.

**Table 4 cancers-12-03826-t004:** Polymorphisms of genes analyzed and disease-free survival (DFS) as well as overall survival in patients with hepatocellular carcinoma.

SNP		*n*	DFS			OS		
			Median DFS (Months) (95% CI)	RR (95% CI)	*p* Value	Median OS (Months) (95% CI)	RR (95% CI)	*p* Value
IL-8					**0.047**			0.366
	AA	14	20 (8.7–40.1)	**2.040 (1.000–4.179)**		22 (8.7–35.3)	0.710 (0.337–1.479)	
	AT ^1^	65	32 (18.3–45.8)	1 (reference)		32 (10.5–53.5)	1 (reference)	
	TT ^1^	46						
VEGF					**0.045**			0.350
	CT ^1^	41	20 (12.0–30.0)	**1.668 (1.007–2.764)**		27 (18.1–36.0)	1.038 (0.646–1.667)	
	TT ^1^	5						
	CC	81	41 (12.5–69.5)	1 (reference)		38 (27.0–50.0)	1 (reference)	
IL-1					0.120			0.229
	CC	54	51 (39.8–61.8)	0.583 (0.321–1.060)		38 (20.5–55.5)	0.657 (0.392–1.1)	
	TT	21	20 (12.1–27.8)	1.060 (0.545–2.063)		38 (0.0–90.0)	0.671 (0.335–1.342)	
	CT	43	25 (17.0–33.0)	1 (reference)		23 (8.0–38.0)	1 (reference)	
IL-6					0.301			0.415
	GG	54	25 (11.8–38.2)	1.552 (0.873–2.759)		32 (16.9–47.1)	0.856 (0.522–1.404)	
	CC	23	32 (1.4–62.7)	1.458 (0.719–2.955)		65 (12.2–117.7)	0.632 (0.318–1.258)	
	GC	50	38 (2.7–73.3)	1 (reference)		30 (14.8–45.2)	1 (reference)	
IL-10					0.580			0.456
	TT	7	21 (6.4–35.6)	1.172 (0.343–4.009)		33 (17.3–47.8)	0.914 (0.271–3.079)	
	GG	74	30 (16.0–44.0)	1.356 (0.763–2.410)		30 (21.0–39.0)	1.359 (0.802–2.301)	
	GT	41	30 (13.0–47.0)	1 (reference)		41 (19.7–46.3)	1 (reference)	
EGFA					0.145			0.073
	AA ^1^	63	24 (0.9–47.1)	1.463 (0.877–2.441)		29 (15.8–42.2)	1.519 (0.959–2.406)	
	GG ^1^	10						
	AG	50	30 (13.2–46.8)	1 (reference)		38 (18.7–57.3)	1 (reference)	
EGFR					0.393			0.523
	GG	50	18 (7.8–28.2)	1.272 (0.721–2.243)		23 (14.0–32.0)	1.177 (0.703–1.972)	
	AA	27	38 (0.0–76.3)	0.807 (0.385–1.690)		38 (0–81.9)	0.819 (0.420–1.597)	
	AG	45	33 (23.1–42.9)	1 (reference)		47 (22.4–71.6)	1 (reference)	
CXCR					0.693			0.847
	GG	29	30 (13.4–46.6)	1.012 (0.527–1.941)		42 (4.8–79.2)	0.836 (0.455–1.538)	
	CC	11	21 (15.5–26.5)	1.414 (0.632–3.162)		30 (22.5–37.5)	0.960 (0.436–2.115)	
	GC	87	35 (17.4–52.6)	1 (reference)		31 (15.8–46.2)	1(reference)	
p53					0.897			0.673
	GG	43	38 (3.1–72.9)	0.653 (0.343–2.000)		38 (25.0–51.0)	0.944 (0.410–2.171)	
	CC	36	24 (7.8–40.2)	0.961 (0.528–1.800)		30 (12.0–48.1)	1.254 (0.687–2.3)	
	GC	16	25 (7.8–40.2)	1 (reference)		35 (15.4–55.0)	1 (reference)	
HIF					0.229			0.779
	CC	7	30 (16.7–43.4)	0.555 (0.135–2.279)		40 (0.0–90.4)	1.214 (0.439–3.355)	
	TT	18	56 (36.1–75.2)	0.518 (0.222–1.209)		23 (6.9–39.1)	1.223 (0.656–2.280)	
	CT	102	26 (18.7–41.3)	1 (reference)		35 (24.1–45.9)	1 (reference)	

^1^ Polymorphisms were grouped together in a codominant model. Results of the regression analyses are given as relative risk with 95% confidence interval. Bold is used to highlight significant results.

**Table 5 cancers-12-03826-t005:** Multivariable analyses of IL-8, VEGF, and DFS.

**SNP**	**Multivariable Analysis**			
	***n***	**Adjusted RR**	**Adjusted *p* Value**	**Median DFS (Months)** **(95% CI)**
IL-8 AA = unfavorable	14	**2.817 (1.278–6.168)**	**0.010**	33 (19.6–44.4)
IL-8 AT/TT = favorable	111	1 (reference)		20 (0.0–40.1)
VEGF TT/CT = unfavorable	46	1.351 (0.736–2.480)	0.332	20 (12.0–28.0)
VEGF CC = favorable	79	1 (reference)		39 (10.2–67.8)
**SNP**	**Combined Analysis**			
	***n***	**Adjusted RR**	**Adjusted *p* Value**	**Median DFS (Months)** **(95% CI)**
0 unfavorable	70	1 (reference)	**0.034**	58 (32.4–83.6)
1 unfavorable	54	1.853 (1.045–3.284)		20 (13.8–26.2)
2 unfavorable	3	4.910 (1.047–23.031)		7 (0.0–10.0)

Adjusted: based on cirrhosis, T-status, and HCC tumor nodes, which were significant in DFS. Results of the regression analyses are given as relative risk with 95% confidence interval. Bold is used to highlight significant results.

**Table 6 cancers-12-03826-t006:** Forward and reverse primer sequences and restriction enzymes used.

No.	Gene	Allele	Primer Forward	Primer Reverse	Annealing Temp.	Enzyme
1	IL-8 251	A > T	TTG TTC TAA CAC CTG CCA CTC T	GGC AAA CCT GAG TCA TCA CA	60 °C	Mfe I
2	VEGF +936	C > T	AGA CTC CGG CGG AAG CAT	TGT ATG TGG GTG GGT GTG TC	60 °C	Nla III
3	EGFR -497	G > A	TGC TGT GAC CCA CTC TGT CT	CCA GAA GGT TGC ACT TGT CC	60 °C	Bstni
4	EGFA 61G	A > G	CAT TTG CAA ACA GAG GCT CA	TGT GAC AGA GCA AGG CAA AG	60 °C	Alu Ia
5	p53	C > G	ATC TAC AGT CCC CCT TGC CG	GCA ACT GAC CGT GCA AGT CA	60 °C	Bstni
6	CXCR1	G > C	CTC ATG AGG ACC CAG GTG AT	GGT TGA GGC AGC TAT GGA GA	60 °C	Alu I
7	IL-1b	C > T	GTT GTC ATC CAG ACT TTG ACC	TTC AGT TCA TAT GGA CCA GA	60 °C	Taq1
8	HIF1 A588T	C > T	CCC AAT GGA TGA TGA CTT CC	AGT GGT GGC ATT AGC AGT AGG	60 °C	Tsp-45 I
9	IL-10	T > G	GAGCACTACCTGACTAGCATATAAG	GTGGGCTAAATATCCTCAAAGT	60 °C	RSAI
10	IL-6	G > C	GCC TCA ATG ACG AC	TCA TGG GAA AAT CC	60 °C	NiaIII

Abbreviations used: VEGF, vascular endothelial growth factor; EGF, epidermal growth factor; EGFR, epidermal growth factor receptor; IL-1/6/8/10, interleukin-1/6/8/10; CXC2, chemokine receptor; HIFa, hypoxia-inducing factor alpha; p53, protooncogene 53.

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
