# Peer review of "Impact of Angiogenesis- and Hypoxia-Associated Polymorphisms on Tumor Recurrence in Patients with Hepatocellular Carcinoma Undergoing Surgical Resection"

_cancers, 2020, doi:10.3390/cancers12123826_

Round 1

Reviewer 1 Report

This study by Miller and colleagues presents novel and interesting data on the impact of SNPs in genes involved in angiogenesis and hypoxia on tumor recurrence in HCC patients undergoing tumor resection. This is an important topic, as liver tumor recurrence after surgery is a major clinical unresolved issue. Mechanistically, angiogenesis, and hypoxia may represent major regulatory pathways involved in tumor recurrence, and therefore the identification of gene variants impacting these processes is very relevant. This study was well conducted and the data are relevant. I only have a few issues for the authors to address which if properly dealt with may strengthen this report.

  1. The authors should justify better how they selected the specific variants that were analyzed in these hypoxia and angiogenesis-related genes. Several other variants have been studied in these same genes concerning HCC patients’ outcome (see for instance PMID: 32020517, the authors should discuss this previous relevant work).

  1. A relatively recent study evaluated the association between IL8 variants and the efficacy of TACE in patients with liver cancer (PMID: 26400525). It would be worth discussing this previous relevant study.

  1. Regarding the VEGF C+936T variant, is there any information regarding its functional impact on the expression/activity of VEGF? Has this variant been related to clinical outcomes in HCC patients?

Minor point

In point 2.4. of the Results, lines 135 and 138, the authors refer to “IL8-254” variants. It should be IL-251 variants.

Reviewer 2 Report

Their study investigates the prognosis of hepatocellular carcinoma patients based on a selection of ten different single nucleotide polymorphisms from angiogenesis carcinogenesis, and hypoxia pathways. Their study with 127 patients found supporting evidence that polymorphisms in angiogenesis-associated pathways correlate with disease-free survival and clinical outcome in patients with hepatocellular carcinoma. They identified SNPs of several genes related to angiogenesis including IL-8. Moreover, they found that IL-8 T-251A (A/A) developed significantly earlier tumor recurrence after partial hepatectomy. Additionally, they suggested high expression of IL-8 T-251A (rs4073) and vascular endothelial growth factor (VEGF C+936T rs3025039 has significant correlation with DFS. I believe this type of data/study is quite informative for subgrouping high risk for recurrence. Therefore, I recommend this paper to be published in Cancers.
